# Modeling a New Supplier Preference Paradigm: A Business-to-Business and African Developing Economy Context

**Rodney Duffett *** and **Myles Wakeham**

Marketing Department, Faculty of Business and Management Sciences, Cape Peninsula University of Technology, Cape Town 8000, South Africa
* Correspondence: duffetr@cput.ac.za; Tel.: +27-021-460-3072

**Abstract:** The recent COVID-19 pandemic, and subsequent invasion of Ukraine by Russia, has demonstrated to the world the volatile and fragile nature of global supply chains. Hence, this study is based on research gaps that propose new sustainable business-to-business (B2B) procurement/supplier models that consider different factors across industries and uses the triple bottom line (TBL) framework as the theoretical underpinning. The study used a quantitative methodological approach and convenience sampling to survey 445 organizations in a B2B procurement context in South Africa. The data were analyzed via structural equation modeling. The inquiry revealed that service quality was important to determine access to personnel and environmental sustainability, which had a favorable influence on personal relationships and gifts and, in turn, positively influenced relationships with salespeople and management. Culture, employment equity, and affirmative action positively affected Black Economic Empowerment status which had a favorable influence on the preference of suppliers' salespeople. Several other positive associations were revealed, which resulted in a unique theoretical supplier preference contribution to the TBL framework. The study also provides organizations with a number of practical benefits stemming from the associations between the new sustainable B2B procurement/supplier constructs that are important as value-added business activities in an African developing economic context.

**Keywords:** business-to-business (B2B) procurement; supply chain management; triple bottom line (environmental, social, and economic); 3Ps (planet, people, and profit); salesperson preference; service quality; management relationships; environment sustainability; Black Economic Empowerment (BEE); employment equity (EE); affirmative action (AA); personal relationships and gifts

## 1. Introduction

Tortato et al. [1] propound that 2020 brought a new global crisis in the form of the COVID-19 pandemic and advance the need for new business frameworks and paradigms in increasingly difficult operating conditions, which require fresh and innovative solutions to reboot the economies of the world. Hence, because of globalization, the COVID-19 pandemic, and more recently, the invasion of Ukraine by Russia, the global supply chain has become more complex, resulting in uncertainty, increased risk, supply chain elongation, and a worldwide shortage of food, fuel, power, and much-needed medication. Furthermore, trillions are spent by companies, organizations, and consumers on an annual basis in South Africa and around the world on services and goods, and organizations and consumers will continue to procure goods and services as the economy starts to regrow over time. Bidmead and Marshall [2], Gregurec et al. [3], and Stalmachova et al. [4] propose that the new norm, in the wake of COVID-19 and ever-changing technology, will continue to impact businesses and disperse workers, allowing them to work remotely, facilitating a bottom-up leadership approach through employee empowerment, and engendering deeper and mutually rewarding relationships, collaboration, and integration within the supply chain. Supply chain management (SCM) is an amalgam of general management, business

processes, and logistics management. All three of these activities have to work in unison to provide the world with the goods and services that it requires in order for it to survive. Logistics consists of a number of activities, of which procurement is a vital activity, since without the requisite inputs such as raw materials, components, spare parts, etc., need-satisfying goods and services cannot be produced [5]. Mensah and Tuo [6], Changalima et al. [7], and Cragg and Chraibi [8] state that procurement is the strategic activity that allows firms to obtain inputs, such as materials and services, so that need-satisfying outputs may be generated for customer consumption and use. Hence, the identification of suppliers, vetting them, and finally the appointment of appropriate suppliers are important logistics activities as the wrong choice could lead to poor quality outputs, reduced sales, reduced profits, poor customer, and consumer service, and as importantly, irreparable reputational damage [9]. Therefore, there is a need to seek a greater understanding of vendor selection criteria and variables that buying organizations employ to assess the services of suppliers, as well as determine the relationships between these variables.

The study's theoretical underpinning is based on Elkington's [10] triple bottom line (TBL) sustainability framework, and its three main pillars, namely, planet, people, and profit (3Ps), which are also commonly referred to as the environmental, social, and economic pillars. Luthra et al. [11] declare that environmental sustainability is an essential obligation to protect worldwide ecosystems and conserve natural resources to support the well-being of people around the globe, both now and in the future. Sustainability is explained as serving current consumer wants and needs without jeopardizing and compromising the capacity of generations in the future to provide for their acquisition desires in terms of the use of natural resources. Organizations should secure services and products that are not harmful to the environment (planet), which should have been sustainably produced. For example, the use of biodegradable and recyclable products and packaging, and services and products that are not responsible for excessive greenhouse gasses due to the utilization of renewable raw material and energy in their production [12–14]. A number of studies propose further research on greener and/or more sustainable environmental approaches [15–27]; hence, organizations should implement and develop sustainable procurement practices, which should include strategies, guidelines, and procedures to assist with more environmentally friendly and sustainable procurement decisions [12,15,16].

Social (people) procurement considers issues to ensure sustainability, such as divergent religious, cultural, and political beliefs, non-heterosexual orientation, the disabled, elderly, and children, previously disadvantaged people (i.e., Black Economic Empowerment (BEE), employment equity (EE), and affirmative action (AA)), the inclusion of women in the workplace, employment creation, and diversity and equality goals [28–30]. BEE aims to provide South African previously disadvantaged (Black, Asian, and Colored) people with equal opportunities that were afforded to White citizens as a mechanism to rectify the inequalities of the past [29,30]. Provision for government tenders was included in the new Constitution and so Black-owned companies are essential for securing government business. This has expanded beyond local and national government and now includes conducting business with medium to large organizations [31,32]. The abolition of apartheid and separate development have pressured organizations to adopt AA and EE and to convert the old nationalist prototype autocratic organizational culture to one of *Ubuntu* (humanity towards others). Leonard and Grobler [33], Horwitz and Jain [34], Jain et al. [35], and Gopalakrishnan and Zhang [36] put forward that the culture of a vendor can either have positive or negative impacts on the vendor's salespeople's attitudes towards their customers that warrants further investigation. A number of studies report that meaningful personal business relationships (social pillar) are advantageous, since the strength of supplier relationships can be used to increase the chance of survival in highly volatile marketplace climates [37–40]. From a relationship point of view, Grewal et al. [41], Pawłowski and Pastuszak [42], and Paesbrugghe et al. [43] suggest that although relationships between the seller and the buyer are important, one needs to differentiate between a working relationship, where the interests of the organizations are uppermost in the minds of the parties, and one which can

be used by the seller to leverage an advantage over the buying firm. Babb et al. [44] and Chen et al. [45] assert that there is a relationship between gifting and the expectation of reciprocity from the giver, which is another social business-to-business (B2B) procurement aspect that requires further investigation.

The sourcing of suitable suppliers is an essential function for all organizations to ensure profitability (economic). Once relationships have been formed with various suppliers, organizations should examine supplier performance via an appropriate appraisal system on a regular basis, which, over time, could aid them in developing a preferred supplier list [46,47]. Kalkanci and Plambeck [47] propose that supplier examination serves as an organizational control measure to identify appropriate vendors with the capacity to produce the necessary services and products. Access to supply personnel and service that exceeds buying company expectations is an important selection criterion nowadays to maintain profit levels. To be involved in the pre-transactional, transactional, and post-sales elements, the buying firm (customer) needs to have access to a variety of supplier personnel other than the salesperson [48]. Client involvement with the successful transaction of orders is of vital importance to both the supplier and B2B customer and whether the needs of the customer are satisfied or not, most customers expect service that is of the highest level and that can exceed their expectations [49]. Lawrence [49] further mentions that alienating the client could expose both the firm and the client in a negative way that could be detrimental to profits.

Hence, this study adopted various procurement/supplier elements that emanate from the 3Ps of the TBL sustainability framework, as well as supply-related factors that are inherent to South Africa with the aim to explore a new B2B supplier preference model. The research explores various associations between procurement factors such as the preference of suppliers' salespeople, BEE status, service quality, relationship with salespeople and management, environmental sustainability, culture, EE, and AA, personal relationships and gifts, access to personnel, and exceeding expectations, which highlights the importance of procurement as a main value-added business activity in a South African context.

A number of research gaps were identified based on the above B2B procurement, supplier, and TBL discourse. Saxena and Seth [23] suggested that further research was needed to investigate TBL substantiality issues. Boruchowitch and Fritz [50] confirm that more research is required based on sustainable SCM and procurement. Epoh and Mafini [21], Nilsson and Göransson [22], Voola [25], Boruchowitch and Fritz [50], Husu [51], and Zhu et al. [52] suggest larger sample sizes and/or different contexts should be employed in future B2B procurement and supplier inquiry. Several studies call for additional studies that consider the sustainable environmental pillar of B2B procurement [21,22,24–26]. Nilsson and Göransson [22], Voola [25], Husu [51], Butt [53], Chang et al. [54], Difrancesco et al. [55], Mangus [56], Santos and Cabral [57], and Zhou [58] stipulate supplementary investigation on the different social and/or relationship aspects of suppliers and B2B buyers. Padgett et al. [59] mandate that relationship investment and service quality require further inquiry in terms of B2B buyer and supplier relationships. Yu [26] also indicates that future research should survey different levels of B2B procurement personnel and not only at management levels (social pillar). A number of inquiries assert that different factors and/or constructs should be considered in the evaluation of suppliers and sustainable B2B procurement [21–23,25,51,55,60,61]. Nilsson and Göransson [22], Voola [25], Difrancesco et al. [55], Cherian and Arun [60], and Mufti and Aprianingsih [62] reveal that various economies and sectors have divergent B2B procurement, supplier, and marketing needs, so further research is necessary across countries, economies, and industries. Manchanda and Deb [63] confirm that B2B procurement studies should consider buyers across multiple industries, which should not be limited to individual industries and sectors. Salam and Bajaba [61] also affirm that additional research on SCM, marketing, and procurement is necessary in developing countries. Qazi and Appolloni [27] postulate that further examination is required to revamp SCM and B2B procurement operations. Mohan [64] reviewed 188 papers and identified a number of important B2B buying research topics that

warrant further research, viz., B2B relationship marketing, organization relationship, and sales and personal selling. Several inquiries propose that in a climate of uncertainty and risk, new B2B procurement models should be developed to mitigate these circumstances, especially in developing countries [63,65–67]. This study endeavors to address a number of the aforementioned research gaps and make an original contribution by conducting additional research that considers B2B procurement, supplier, and the TBL framework elements from a developing country perspective. Hence, there is a need for this study, which is justified via literature, since it investigates new sustainable B2B procurement environmental [21,22,24–26], social [22,25,51,53–59], and economical factors [23,50] and/or constructs [21–23,25,51,55,60,61,64], uses a large sample size [21,22,25,50–52], and seeks to develop a new supplier preference model [27,63,65–67] in an African developing country across various industries [22,25,55,60–63]. To determine the aforementioned points, the researcher believed that instead of being prescriptive (that is, looking at the problem from a seller's point of view) this study would determine the variables from buyers who procure goods and services on behalf of their respective organizations. Hence, the main research question of the study is:

> "What are the associations between the preference of suppliers' salespeople, BEE status; service quality; relationship; environment sustainability; culture, EE, and AA; personal relationships, the provision of gifts; access to management and exceeding expectations in terms of a new B2B supplier preference model among organizations in South Africa?"

The necessity for this study, which considers the aforementioned B2B supplier preference variables, is also justified via literature since a number of authors conducted research on the preference of suppliers' salespeople [5,46,47], BEE status [31,32,34], service quality [9,59], relationship with salespeople and management [37–40], environment sustainability [11–27], culture, EE, and AA [28,33–36], personal relationships and the provision of gifts [41–45], and access to management and exceeding expectations [48,49]. In summary, the inquiry seeks to understand what bases organizational buyers select their vendors so that sales organizations (vendors) can better align their services to satisfy B2B procurement vendor needs via a new B2B supplier preference paradigm that used the 3Ps of the TBL framework as the foundation of the proposed model.

## 2. Literature Review and Hypotheses

### 2.1. Procurement

Procurement is described as the acquisition process to secure services and goods to ensure efficient logistics and manufacturing operations and processes [68]. This function includes the identification of suppliers, their assessment and selection, vendor audits (to establish vendor effectiveness and efficiency in meeting organizational requirements), quality control, price determination, purchase timing, and material determination. Kristensen et al. [69] and Anin et al. [70] propose that the procurement objectives of organizations are to ensure the efficient running of an enterprise, to competitively purchase services and goods of a sufficient standard, and to economically secure the greatest monetary value. Harrison et al. [71] posit that procurement and logistics are essential business performance functions, so if an organization can enhance its logistics activities, it can create a competitive advantage. Renukappa et al. [72] and Alabdali et al. [73] propose that a competitive advantage leads to a healthy profit and the sustainability of organizations is dependent on recurring revenue and growth. Organizations spend much of their money securing materials and goods that are required to maintain business operations, so strategic procurement is essential for an organization's success. Hence, organizations must establish a procurement guide to outline best practices for purchase decisions. It is important to have a strategy that vets appropriate vendors and appoints them as important supply chain partners. Décary-Hétu and Quessy-Doré [9] state that vendors need to be vetted based on various criteria such as service quality, price, and financial stability. The selection of effective and efficient suppliers has a huge impact on cost, since many organizations

spend up to half of their income to secure services, goods, and raw materials from external vendors. After the identification of cost factors, plans can be implemented to address the unique circumstances, especially in South Africa, as shown in the ensuing supplier preference variables, and establish and maintain relationships with suppliers over the long term.

More recently, researchers and authors such as Omurca [74], Sultana et al. [75], Stević [76], and Noshad and Awasthi [77] advocate that besides the comprehensive list of criteria, other parameters such as relationships, mutual-dependence, the profitability of the vendor's offering, the proximity of the vendor's operation facilities, and the vendor's use of procurement technology all play important roles in terms of vendor appointment and support. Kennedy and Deeter-Schmelz [78] maintain that in B2B marketing, the future salesforce role is unclear in terms of the growth of online purchases, hence the reason it is important for organizations to not only know what they are purchasing but from who they are buying, as the vital link between the organizations (salespeople) could be replaced by remote salespeople. This is concurred by Ulaga and Kohli [79] who submit that the role of a traditional salesperson will morph into a solutions salesperson that will focus on reducing uncertainty and fostering adaptiveness. Olorunniwo and Jolayemi [80] indicate that there are no clear suitable selection criteria to select vendors in various divergent scenarios, especially in South Africa. Therefore, this study analyzes the relationships between various procurement selection criteria measures to provide further insight into supplier preference and salespeople across different types and sizes of South African organizations.

### 2.2. Hypotheses

Muyeed [81] suggests that service quality is essential in a B2B customer's perception of the total offering and will be the dominant parameter in evaluating customer expectations and satisfaction. Shirouyehzad et al. [82] posit that service quality is the competence of a supplier to effectively service customer needs in terms of their expected requirements and is vital in the selection process and retention of the supplier's services. Hassan [5] postulates that the purchasing function is a vital aspect of logistics management, and the main responsibility of this function is the selection of the most efficiently performing vendors. Several essential criteria are considered in evaluating vendors' relative efficiencies, namely price, product, and service quality. Abdolshah [83], Mishra et al. [84], Taherdoost and Brard [85], and Zygiaris et al. [86] submit that quality products and services are the main criteria for supplier evaluation and selection, and further posit that service of a high-quality results in customer loyalty and satisfaction to the provider of the service. Researchers suggest that a high service quality standard results in customer loyalty and satisfaction among the organization's consumers [87]. Griffith and Lee [88] confirm that the relationship between having access to customer personnel cannot be denied and that likewise, its relationship with service excellence cannot be argued. It makes logical sense that the provision of service excellence (which exceeds expectations) will assist supplier organizations to have good relations with B2B customers and their employees, and that such outstanding service will help ensure access to personnel, centers of influence, and senior management. Hence, it is hypothesized that:

**Hypothesis 1 (H1).** *Service quality has a favorable impact on access to personnel and exceeding expectations.*

Service quality can have an important influence on environmental sustainability, besides service quality having a positive influence on access to B2B customer personnel and exceeding customer expectations. For a buying organization (a B2B customer) to offer sustainable offerings, the inputs facilitated by their suppliers and their suppliers' vendors need to be sustainable. Kaswan et al. [17], Rathi et al. [18,19], and Yadav et al. [20] reveal that productivity-related criteria, management commitment, and the availability of finance were the most important factors in the implementation of TBL sustainability via a green

lean six sigma approach. Miller et al. [89] reveal that organizations must consider the larger social, political, and environmental climate, as well as suppliers, customers, investors, and government roles. To do this, organizations need to have a sustainable business strategy, which should include the 3Ps. Bonn and Fisher [90] proclaim that the matter of sustainability is a strategic factor for organizations in uncertain times, and that often it is the missing link in corporate strategy. Researchers advocate that businesses and their management functional areas have been pressurized at global, regional, and local levels to consider sustainability (in terms of environmental, social, and economic issues) in all decisions used to increase the competitive advantages of organizations [91–93]. This starts with sourcing suppliers and their inputs, then the employment of sustainable production processes through to providing the final service and product offering to the consumers. Thus, it is hypothesized that:

**Hypothesis 2 (H2).** *Service quality has a favorable impact on environmental sustainability.*

Previous studies show that divergent organizational culture can adversely affect relationship satisfaction, financial performance, and productivity between organizations [94,95]. Cultural similarity is a good indicator of favorable outcomes between two organizations [96]. Thus, this study posits that a cultural fit between two organizations, the provision of superior service, the forging of strong mutually rewarding relationships, and free access to a buying organization's employees can have a favorable effect on the culture of an organizational customer. This inquiry further postulates that access to buying organizations' employees by vendors who practice employee-centric EE and who exceed customer expectations will have a positive influence on the customer. The rationale is that if the supplier adopts EE-centrism as a policy and treats its employees with the dignity that they deserve, there should be a positive spill-over when vendor marketing employees communicate with the buying organization personnel. In other words, if the vendor has a rich culture in EE, and employs previously disadvantaged operational employees and salespeople (AA policies), it should, through the behavior of the vendor's salespeople and personnel, have a positive impact on the buying organization [97]. Thus, it is hypothesized that:

**Hypothesis 3 (H3).** *Access to personnel and exceeding expectations has a favorable impact on culture, EE, and AA.*

Environmental sustainability is often explained as fulfilling current wants and needs without harming future cohorts to satisfy their acquisition and use of natural resources desires [11]. Unlike AA and EE, there is currently no enforceable legislation to pressure South African firms to adopt sustainability, although this could change in the future if organizations ignore the people and planet sustainability pillars in pursuit of profit. The primary goal is to establish supply chain practices that are sustainable to create a smooth flow through the entire supply network, and to achieve this they must establish long-term sustainability goals [98]. So, with the above in mind, should both institutions (buyer and seller) practice environmental sustainability for their mutual benefit and the benefit of their stakeholders, closer relationships will be forged as a result of such compliance. In other words, in this instance, environmental sustainability will have a significant and beneficial influence on relationships. Several investigations agree that firms should look at green supply chain selection criteria [11,17–27,99–101]. Customers have increased their knowledge, and markets and various stakeholders place ecological pressures on organizations to develop strong relationships with supply chain members who green-up [11]. The provision of gifts can lead to a conflict of interest if the gift receivers sacrifice the organization's interests to receive personal gains at the expense of the employer but can foster relationships [102]. For example, customers may reciprocate with loyalty to the gift-giver organization, but the value of the gifts should be small. Alikhani [103] avouches that the selection of vendors is a multi-stage and complex strategy; few studies consider

both risk management and sustainability issues concurrently, and risk can be diminished by mutually beneficial relationships. Therefore, it is hypothesized that:

**Hypothesis 4 (H4).** *Environmental sustainability has a favorable influence on personal relationships and gifts in the new B2B supplier preference model.*

Mocke et al. [104] emphasize that personal relationships developed among parties that are mutually connected through business and have a collaborative relationship with one another can result in interdependent associations. Personal relationships can benefit buying and selling organizations and their employees via more effective communication, knowledge of each entity's business processes and offerings, improved problem-solving, reduced costs, continuous operational improvements, shared processes and systems, and mutual trust [105]. Gligor and Holcomb [106] affirm that personal relationships can often be separated from other business-oriented relationships since these may result in personal relationships outside of a work environment. Professional interests may improve, and friendships can be developed owing to enduring business-related personal relationships [107]. Gunduz and Önder [108] express that while giving gifts is normally a simple, voluntary, and kind gesture, it often creates obligations to give something in return. Hence, the value of a gift in a corporate setting is its ability to foster a long-term relationship as opposed to the actual value of the gift. The following is therefore hypothesized:

**Hypothesis 5 (H5).** *Personal relationships and gifts have a significant positive influence on relationship with salespeople and management in the new B2B supplier preference model.*

As previously asserted, the main purpose of BEE is to remedy apartheid's legacy and stimulate previously disadvantaged people's participation in the economy of South Africa. Therefore, to practice BEE, suppliers need to have a culture of employee development, believe in EE as a method to cure the curse of apartheid and the job discrimination that accompanied it in favor of White people, and be steadfast in terms of AA [109]. AA seeks to improve educational and employment opportunities for people that were previously discriminated against due to apartheid, and endeavors to assist in closing the divide between the haves and have-nots. AA includes not discriminating in the workplace regarding identity, disability, gender, sexual orientation, ethnic identity, age, and population group (race) [110,111]. Therefore, if the supplier is acknowledged to be a BEE-accredited supplier, such a supplier should have an embedded culture that reflects its commitment to BEE. Likewise, the EE Act is designed to ensure that employees receive amicable treatment and equal opportunities in the workplace [112]. The law was formulated to negate any form of discrimination and unfair treatment of employees irrespective of race, creed, or religion [113]. Suppliers who practice EE should reflect, by their acceptance and practice of EE that they have embraced BEE, which, as posited above, is to promote the economic engagement of all South Africans and particularly those who were previously discriminated against. Dass and Parker [114] suggest that firms should encourage employees to view AA not as a barrier, but that it should be made a part of customer, vendor, and employee programs. All of the aforementioned considerations will have a positive effect on the BEE status of the supplier, which could be an important selection criterion for many purchasing firms (including large enterprises and government bodies). In terms of the above, the following hypothesis was proposed:

**Hypothesis 6 (H6).** *Culture, EE, and AA has a significant positive effect on BEE status in the new B2B supplier preference model.*

The cementing of close marketing relationships is paramount for survival and growth in the challenging twenty-first-century business milieu [115]. Several studies that propose relationships (social pillar) are imperative in the selection and supporting of suppliers and their sales staff, since the interface between the selling organization and the B2B and

business-to-consumer (B2C) salespeople need to see their relationship with their customers as actual relationships and not focus on transactions [22,25,51,53–58,116–118]. In other words, their goal should be to focus beyond sales transactions and develop beneficial relationships with their customers over the long term so that there are benefits for all of the stakeholders in the relationship. To do so, salespeople need to formulate and maintain strong relationships with customer procurers, management, and staff, and use such associations to influence the aforementioned B2B customer and employees to motivate them to continuously support the vendor and its salesperson. In other words, to use such affinity to create interest and desire to garner enduring support [119–121]. Therefore, the following hypothesis was suggested:

**Hypothesis 7 (H7).** *Relationship with salespeople and management has a significant favorable influence on preference of suppliers' salespeople in the new B2B supplier preference model.*

As previously asserted, the BEE status of a supplier is important, particularly when the customer is a large national and/or international organization, and particularly the government [34,122]. Organizations receive points that contribute to their BEE compliance certificate. Higher levels of BEE compliance increase the prospect of securing government tenders. The Department of Trade and Industry (DTI) [123] reports that organizations that are BEE compliant and have high BEE scores not only benefit by being allowed to supply large companies, but also facilitate attractive tax benefits, and increase their prospects of securing more government business in South Africa [124]. Therefore, the following hypothesis was considered:

**Hypothesis 8 (H8).** *BEE status has a significant favorable influence on preference of suppliers' salespeople in the new B2B supplier preference model.*

In relation to the above, the conceptual model of the B2B supplier preference factors that will be tested is illustrated in Figure 1.

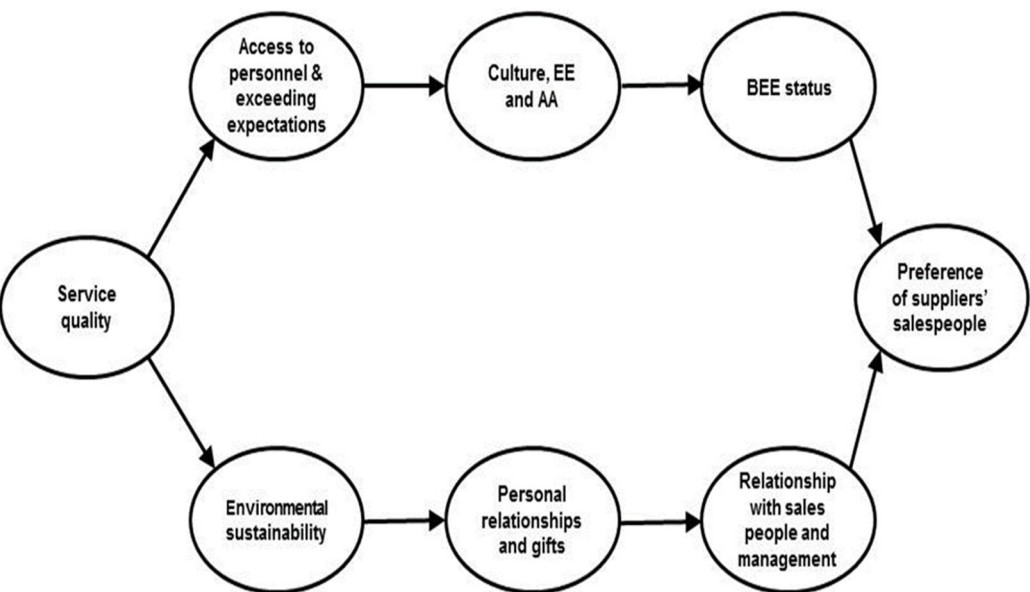

**Figure 1.** Conceptual model of the B2B supplier preference factors.

## 3. Materials and Methods

The research design is described as exploratory and descriptive since this study observed and described the behavior of the research population without influencing it in any way, uncovered ideas and insights, determined relationships between various variables, and established the frequency with which something occurs [125]. The study adopted

the quantitative approach to garner the requisite data via a structured questionnaire to survey small-, medium-, and large-sized organizations operating in South Africa. Several factors of the research onion are used to provide an overview of the study's methodology in Table 1 [126,127].

**Table 1.** Methodology overview.

| Process | Application |
| --- | --- |
| Research philosophy | Positivism |
| Research strategy | Quantitative research |
| Research design | Exploratory and descriptive |
| Time horizon | Cross-sectional |
| Data collection techniques and research methods | Self-completion: electronic and paper (self-administrated) |
| Sample design | Purposive and convenience sampling |
| Measurement | Structured attitude measurements (Likert scales) |
| Data manipulation | Statistical analysis: SEM and hypothesis testing |

### 3.1. Sampling and Collection of Data

The data were collected via purposive and convenience sampling by utilizing the researchers' personal business networks and organizations such as the Chartered Institute of Logistics and Transport, Institute of Purchasing and Supply SA, and Smart Procurement to secure the lists of their members so that the questionnaire could be disseminated to them. As there is no comprehensive list of organizations, the researcher over-estimated the number of businesses in South Africa, and based on a 95% confidence internal level and 5% error margin, a sample size of at least 385 respondents was determined. The questionnaire was distributed via several methodologies, mostly via email and physically (i.e., face-to-face). The data that was collected was coded, captured, and analyzed via statistical software known as SPSS. All completed surveys were thoroughly inspected to see whether they were correctly and fully completed, and those that were incomplete and contained errors were discarded. In this manner, the researcher managed to secure the support of 445 procurement personnel and other suitably qualified (CEO, MD, accountant, and finance manager) respondents.

### 3.2. Description of Sample

The research population comprised different-sized organizations operating in South Africa, within a myriad of business types and sectors. The various business procurement variables associated with the study ranged from the size of the business to whether the organizations had well-defined procurement procedures, strategies, and policies that are outlined in Table 2.

Table 2 shows that organization size was almost identical in number, but large organizations represented more than a third of the total organizations. A large percentage of the respondents worked at local and national levels. The results show that procurement was mainly conducted on a centralized basis, and a large proportion of the respondents were limited to procuring goods up to the value of R50,000 without having to seek approval from higher management. From a procurement strategy, policy, and procedural viewpoint, of the 445 organizations that participated in the research study, almost seven out of ten had formal procurement strategies and policies, whereas nearly three-quarters had procurement procedures.

**Table 2.** Business procurement descriptive statistics.

| Business Procurement Variables | | Freq. | % |
|---|---|---|---|
| Size of business | Small (less than 20 employees) | 140 | 31.5 |
| | Medium (21–200 employees) | 141 | 31.7 |
| | Large (more 200 employees) | 164 | 36.9 |
| Locale of business | Local organization | 196 | 44.0 |
| | National organization | 147 | 33.0 |
| | International organization | 102 | 22.9 |
| Basis of procurement | Centralized purchasing | 348 | 78.2 |
| | Decentralized purchasing | 97 | 21.8 |
| Employment position | Buyer/procurer | 159 | 35.7 |
| | Senior procurement officer/buyer | 52 | 11.7 |
| | Manager: buyer/procurement | 103 | 23.1 |
| | Accountant | 27 | 6.1 |
| | Finance manager | 22 | 4.9 |
| | CEO or MD | 14 | 3.1 |
| | Business owner | 68 | 15.3 |
| Decision level | R50,000 or less | 211 | 47.4 |
| | R50,001 to R100,000 | 70 | 15.7 |
| | R100,001 to R25,000 | 28 | 6.3 |
| | R250,001 to R500,000 | 50 | 11.2 |
| | R500,001 to R750,000 | 11 | 2.5 |
| | R750,001 to R1 million. | 18 | 4.0 |
| | Above R1 million | 57 | 12.8 |
| Procurement strategies | Yes | 308 | 69.2 |
| | No | 137 | 30.8 |
| Procurement policies | Yes | 307 | 69.0 |
| | No | 138 | 31.0 |
| Procurement procedures | Yes | 324 | 72.8 |
| | No | 121 | 27.2 |

*3.3. Measures*

As previously mentioned, the research adopted a quantitative approach, and a structured questionnaire was utilized to survey large, medium, and small organizations in South Africa. A pilot study of ten respondents was used to test the questionnaire and five procurement specialists were consulted to ensure the robustness of the research instrument. The questionnaire used multiple-response questions to gather a wide array of business procurement data from the respondents, namely: size of business, locale of business, basis of procurement, procurement experience, employment level, decision level, procurement strategies, procurement policies, and procurement procedures. The questionnaire, which included ranking and Likert scale-type questions, was structured to assist the researcher to avoid having respondents select a single column throughout the five-point Likert scales. The B2B supplier preference constructs were developed and adapted from a number of authors, viz.: preference of suppliers' salespeople [128–130]; BEE status [29,131–134]; service quality [82,87,135]; relationship with salespeople and management [119,136,137]; environmental sustainability [90,138,139]; culture, EE, and AA [33,34,97,109]; personal relationships and gifts [106,108,140,141]; and access to personnel and exceeding expectations [119,120,142]. Refer to Table 3 and Appendix A.

**Table 3.** Exploratory component analysis—supplier preference.

| B2B Supplier Preference Variables | FLs | AVE | CR | Cronbach's $\alpha$ |
|---|---|---|---|---|
| Preference of suppliers' salespeople (PSS) | | | | |
| PSS1 | 0.668 | | | |
| PSS2 | 0.790 | | | |
| PSS3 | 0.800 | | | |
| PSS4 | 0.699 | 0.553 | 0.908 | 0.881 |
| PSS5 | 0.802 | | | |
| PSS6 | 0.791 | | | |
| PSS7 | 0.741 | | | |
| PSS8 | 0.637 | | | |
| BEE status (BEES) | | | | |
| BEES1 | 0.836 | | | |
| BEES2 | 0.919 | 0.786 | 0.916 | 0.859 |
| BEES3 | 0.902 | | | |
| Service quality (SQ) | | | | |
| SQ1 | 0.825 | | | |
| SQ2 | 0.864 | 0.688 | 0.869 | 0.861 |
| SQ3 | 0.798 | | | |
| Relationship with salespeople and management (RSM) | | | | |
| RSM1 | 0.649 | | | |
| RSM2 | 0.764 | 0.539 | 0.777 | 0.677 |
| RSM3 | 0.783 | | | |
| Environmental sustainability (ES) | | | | |
| ES1 | 0.697 | | | |
| ES2 | 0.888 | 0.653 | 0.848 | 0.761 |
| ES3 | 0.827 | | | |
| Culture, EE, and AA (CEEAA) | | | | |
| CEEAA1 | 0.579 | | | |
| CEEAA2 | 0.872 | 0.494 | 0.748 | 0.716 |
| CEEAA3 | 0.568 | | | |
| CEEAA4 | 0.621 | | | |
| Personal relationships and gifts (PRG) | | | | |
| PRG1 | 0.742 | | | |
| PRG2 | 0.832 | 0646 | 0.845 | 0.823 |
| PRG3 | 0.834 | | | |
| Access to personnel and exceeding expectations (APEE) | | | | |
| APEE1 | 0.806 | 0.644 | 0.784 | 0.785 |
| APEE2 | 0.800 | | | |

The Likert scale statements were manipulated using SPSS (version 25); prior to that Cronbach's alpha ($\alpha$) and convergent reliability (CR) statistical technique was utilized to ascertain the participant response reliability derived from the scales. The convergent and discriminant validity was also analyzed via AVE (average various extracted), factor loadings (FLs), and Fornell and Larcker's [143] formulae (refer to Table 3 and Appendix A).

## 4. Results

### 4.1. Measurement Model

Structural equation modeling (SEM) was utilized to determine the more complex relationships in terms of the suppliers' preference constructs. The SEM used the standardized beta coefficient measures to ascertain the robustness of the associations between the B2B supplier preference variables via Amos (version 23), and exploratory component examination was executed to empirically appraise suppliers' preference variables in terms of

validity and reliability. The sampling suitability was considered via Kaiser–Meyer–Olkin's formula, which yielded a good value of 0.806. The correlation matrix factorability was considered through Bartlett's sphericity test, which ascertained that the different Likert scale correlations were sufficient, since the test was significant at $p < 0.001$ [144,145]. The exploratory component analysis generated eight components (factors) with eigenvalues greater than one. The explained variance for the eight factors was 5.440%, 4.039%, 3.176%, 2.226%, 1.620%, 1.472%, 1.062%, and 1.016% respectively. The total sum of factors explained 69.142% of the variance which revealed a high correlation in the exploratory factor analysis.

Field [144] asserted that FLs (in the pattern matrix) that are larger than 0.5 should be kept. The pattern matrix shows eight factors, which comprise two or more Likert scale items, and FLs of more than 0.5, which were retained. The reliability of the supplier construct was assessed via CR and Cronbach's $\alpha$, which indicated values of 0.748–0.916 and 0.677–0.881, respectively, that are reflective of robust to excellent reliability. Convergent validity values (calculated via AVE and FLs for the suppliers' preference values) were 0.568–0.919 and 0.494–0.786, respectively. All of the corresponding factors' loading values exceeded 0.5 (and were therefore acceptable), except for the culture, EE, and AA AVE value that was only marginally under 0.5 and so retained (refer to Table 3).

Fornell and Larcker [143] formulae showed acceptable discriminant validity since the AVE square root for each of the suppliers' preference constructs exceeded the correlation values of each construct (as seen in Table 4). The SEM analysis, regarding the goodness-of-fit measurement statistics, showed an adequate model: $\chi^2/df$ (1.655); GFI (0.937); CFI (0.969); TLI (0.967); RMSEA (0.038); NFI (0.931); and SRMR (0.076). The $\chi^2$ measure established that the constrained and unconstrained CMF models were significantly different ($p < 0.001$), so the unconstrained CMF model was adopted due to the shared variance.

**Table 4.** Component correlation matrix.

| | | | | | | | | |
|---|---|---|---|---|---|---|---|---|
| PSS | 0.743 | | | | | | | |
| BEES | 0.005 | 0.886 | | | | | | |
| SQ | −0.015 | −0.037 | 0.830 | | | | | |
| RSM | 0.244 | −0.078 | 0.067 | 0.804 | | | | |
| ES | 0.028 | 0.290 | 0.119 | 0.089 | 0.808 | | | |
| CEEAA | 0.109 | 0.346 | 0.001 | −0.058 | 0.414 | 0.703 | | |
| PRG | 0.242 | −0.125 | 0.089 | 0.093 | 0.111 | 0.250 | 0.734 | |
| APEE | 0.060 | 0.088 | 0.216 | 0.081 | 0.150 | 0.144 | 0.249 | 0.803 |

The SEM model independent variables were evaluated via a multiple variable collinearity to test the B2B supplier preference constructs for excessive levels of correlation. The B2B supplier preference constructs tolerance ranged between 0.702 and 0.961 (greater than 0.1) and VIF was 1.070–1.425 (less than 3), which is indicative that the constructs are not excessively correlated. Refer to Table 5 for the construct collinearity statistics.

**Table 5.** B2B Supplier preference constructs collinearity statistics.

| B2B Supplier Preference Constructs | Tolerance | VIF |
|---|---|---|
| BEES | 0.795 | 1.259 |
| SQ | 0.934 | 1.070 |
| RSM | 0.961 | 1.040 |
| ES | 0.772 | 1.295 |
| CEEAA | 0.702 | 1.425 |
| PRG | 0.830 | 1.205 |
| APEE | 0.877 | 1.140 |

*4.2. Hypothesis Testing*

The path coefficients reveal that service quality has a significant favorable positive impact on access to personnel and exceeding expectations (β = 0610, $p < 0.001$) and en-

vironmental sustainability ($\beta$ = 0.223, $p < 0.001$), which supports H1 and H2. The path coefficients show that access to personnel and exceeding expectations has a significant positive effect on culture, EE, and AA ($\beta$ = 0.174, $p < 0.05$), and environmental sustainability has a significant favorable influence on personal relationships and gifts ($\beta$ = 0.216, $p < 0.001$), which supports H3 and H4. The path coefficients reveal that personal relationships and gifts have a significant positive influence on relationship with salespeople and management ($\beta$ = 0.544, $p < 0.001$), and that culture, EE, and AA has a significant positive effect on BEE status ($\beta$ = 0.549, $p < 0.001$), which supports H5 and H6. The path coefficients show that the relationship with salespeople and management ($\beta$ = 0.340, $p < 0.001$) and BEE status ($\beta$ = 0.095, $p < 0.001$) has a significant favorable influence on preference of suppliers' salespeople, which supports H7 and H8. Additionally, there was good variance ($R^2$) for several of the different B2B supplier preference constructs, including access to personnel and exceeding expectations (37.2%), relationship with salespeople and management (29.6%), BEE status (30.1%), and preference of suppliers' salespeople (12.4%), whereas there was low $R^2$ for environmental sustainability (5%), personal relationships and gifts (4.7%), and culture, EE, and AA (3%). Refer to Figure 2 for a graphical illustration of the above-mentioned hypothesis support and $R^2$.

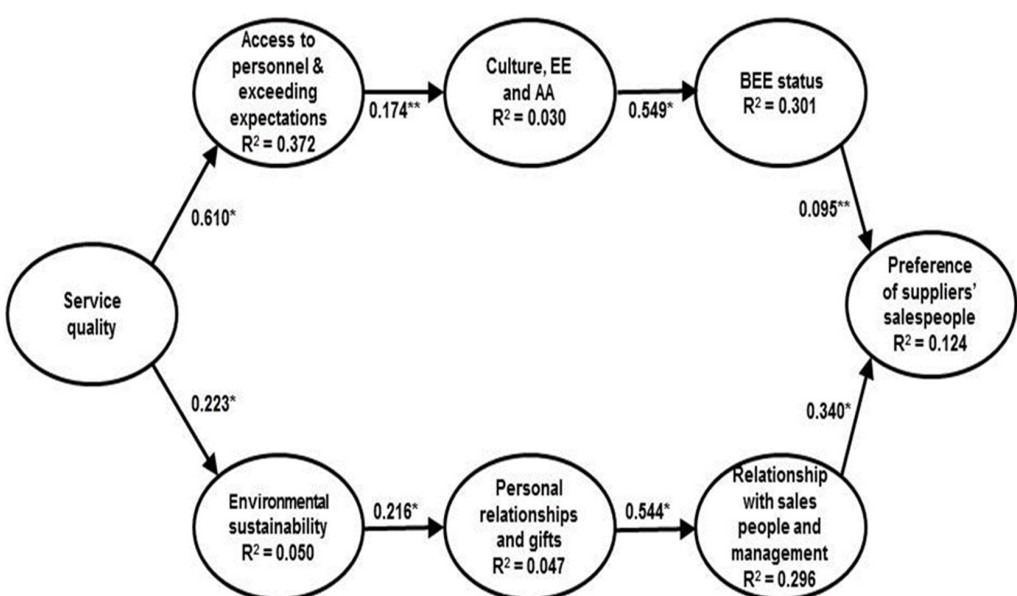

**Figure 2.** New B2B supplier preference model. * $p < 0.001$, ** $p < 0.05$.

## 5. Discussion

Patel and Thakar [146] propose that vendor selection is an important decision-making factor in an organization because of its strategic nature which is multifaceted and includes opposing objectives. Xiao and Kumar [147] suggest that there is a link between service experience and quality since quality service has a huge impact on the relationship between the buyer and the supplier. The delivery of service quality can increase customer support, boost sales, reduce costs, enhance profits, and as importantly, assist in the cementing of long-term and mutually beneficial relationships between selling and buying organizations. Hence, the findings of this inquiry substantiate the aforementioned discourse since it verified a favorable association between service quality, exceeding customer expectations, and access to vendor personnel and exceeding expectations. Exceeding B2B customers' expectations can inspire customers to increase their support, and customers whose expectations are high about a vendor's services or products will not only become brand/vendor loyal, but will pursue an enduring mutually beneficial relationship, thereby helping to increase profits and attain or even exceed revenue objectives for both parties in the equation (thereby realizing the economic pillar of the TBL framework). Several inquiries affirm that supplier evaluation and selection frequently use the quality of the products and services as primary selection

benchmarks [83–87]. Therefore, this study shows that an excellent quality of service will ensure access to the vendors' personnel in an African developing country context, which adds value to the request for further research across economies and countries that have divergent B2B supplier, procurement, and marketing needs [22,25,55,60,62].

Siminică et al. [148] advance that the development of sustainable marketplaces is important to ensure enduring growth and sustainable innovation that should be fostered and encouraged. Sustainable procurement can stimulate ethical practice and fair trade that could facilitate additional investment in African developing economies [149]. Luthra et al. [11], Jain and Singh [150], and Takalo and Tooranloo [151] affirm that the identification of savvy sustainable suppliers is a key aspect of the supplier selection process and service quality. Therefore, the results of this inquiry add value to the abovementioned sentiment as it confirmed a positive significant association between service quality and environmental sustainability in terms of vendor selection criteria, which fulfill Saxena and Seth [23] and Boruchowitch and Fritz's [50] recommendations for additional inquiry to consider sustainable SCM, procurement, and TBL substantiality issues. Miller et al. [89] affirm that strategic and sustainable business goals should have a positive influence on society and the environment, but also be of benefit to the stakeholders. Sustainability starts with sourcing appropriate suppliers and their green inputs, and through a sustainable production process, the outputs will likewise be sustainable. To achieve this, vendors need to market quality products (materials, components, parts, etc.) and services that are sustainable over the long term. Wang et al. [152] suggest that by working closer with suppliers, a buying organization can lessen its environment and social impact and position itself for strong growth and sustainability in supply chains. Therefore, this study helps address Epoh and Mafini [21], Nilsson and Göransson [22], Sharma [24], Voola [25], and Yu's [26] request for additional studies that consider the sustainable environmental aspect of B2B procurement by confirming that service quality positively influences environmental sustainability in a South African B2B procurement environment.

Van Rensburg [153] advocates that by developing skills and EE, business effectiveness will improve due to increased employees' skills. Culture and the practice of EE and AA by the vendor are significant vendor selection measures. Hartnell et al. [96] voice that cultural similarity between organizations is a vital predictor of favorable results such as enhanced relationships and business. It is reasonable to assume that an EE-centric vendor whose employees have access to a B2B customer's employees will, through their behavior, have a positive influence on the customer and its staff. AA sets procedures designed to negate unlawful and unfair discrimination among applicants for employment and education [154]. Firms that practice AA (also known as "positive discrimination") help minorities and disadvantaged people and groups by employing them as a priority and assisting them to find housing and gain admission to universities and colleges. Hence, this study makes a unique contribution as it revealed a positive significant relationship between personnel and exceeding expectations and culture, EE, and AA. If a vendor practices AA in its workplace, like culture and EE, by employing previously disadvantaged operational employees and salespeople, and uses them to access buying firms' personnel and exceed their expectations, both organizations will receive social and economic benefits.

Villena and Gioia [98] found that an increasing number of international organizations are committed to collaborating with vendors that observe environmental and social standards. These researchers also proclaim that organizations should work towards creating a more sustainable supply chain. Saad et al. [155] state that strategic relationship management is the maintenance and development of engagement that is sustainable and continual by an organization with its customers and suppliers. Accordingly, the current inquiry makes an original contribution since it proved a significant relationship between environmental sustainability and personal relationships and gifts, which assists in narrowing the knowledge gap of inquiries that mandated further investigation on sustainable environmental procurement practices and decision-making [17–27]. The formulation of green procurement tactics needs full organizational commitment to establish green objectives

and implement these in a sustainable manner [156]. Hence, setting and observing sustainable green procurement specifications and other interventions are not easy tasks [157,158]. Therefore, B2B buyers and sellers should practice environmental sustainability for their mutual benefit and the benefit of their stakeholders, so that closer relationships will be forged as a result of such compliance. Although gifting and the forging of close personal relationships are frowned upon in many business sectors, it is difficult for suppliers to inculcate customer relationship management (CRM) without befriending their B2B customer employees. From a sustainability perspective, should the gift receiver (customer) be green-procurement focused, and insist, where possible, that the vendor markets sustainable offerings, then the giver (vendor) should respond as such and focus on the provision of green products and services, which will enhance the relationship between the two entities. Few studies have considered the association between environmental sustainability and personal relationships and gifts in a B2B procurement context [159,160], so this study adds new knowledge to the TBL framework in this regard.

Salespeople who are engaged and connected to their customers are employees who provide superior customer service, value, and a unique customer experience, which translates into customers who are prepared to support the suppliers' salespeople, their management team, and the selling organizations [161]. This is a notion supported by Cadwaller et al. [162], Hollebek [163], and van Doorn [164] who assert that customer engagement by supplier salespeople is an important requisite for relationship building, service excellence, and continued customer support. Gable and Reis [107] postulate that the professional interests of organizations can be improved by establishing personal relationships and building friendships that can be enduring in nature. Hence, this research adds value to the social pillar of the TBL framework since it found a significant relationship between personal relationships and gifts and relationships with salespeople and management. Furthermore, the results help fulfill the gap in knowledge that proposes additional inquiry on the social and/or relationship elements of B2B procurers and suppliers [22,25,51,53,54,56–58]. Gifts are common in a B2B environment to establish and sustain relationships between business customers and suppliers but frequently result in ethical problems in sales, purchases, and SCM [165]. This study shows that the relationship between the vendor's employees and that of the customer can be enhanced, as stated above, by the gifting of small and low-value gifts and a personal relationship [166], which is characterized by mutual commitment, collaboration, trust, and friendship, in the provision of long-term economic benefits for both the seller and the buyer.

BEE is a unification South African government program to remedy past injustices of separate development (apartheid) [3]. Shava [30] and Forbes et al. [167] affirm that the key reason for BEE was to ensure advancement in economic participation and transformation for all Black individuals in the economy of South Africa. BEE is a significant selection criterion and Black-owned organizations are indispensable to secure government contracts and tenders, especially for the vendor's second-tier suppliers and their third-tier suppliers. Therefore, BEE status is also salient in the consideration of supplier appointment [31,32]. This research adds significant value to B2B procurement and supplier discourse, since it affirmed a significant relationship between culture, EE, and AA and BEE status, and so contributes to the limited discourse that considers these inherently unique South African social and economic B2B procurement factors. Therefore, this study heeds the call for additional research to consider divergent factors and/or constructs in the evaluation of sustainable B2B procurement and suppliers [21–23,25,51,55,60,61]. Business culture includes procedural and behavioral norms that consist of a code of conduct, policies, employee attitudes and behaviors, goals, values, ethics, and procedures. EE, AA, and BEE are therefore seen as part of organizational culture as they impact the behavior, values, and attitudes of employees. Suppliers who practice EE and AA are perceived to have embraced BEE, which has a positive effect on the BEE status of the supplier and could be an important selection criterion for many purchasing firms that include large enterprises and government bodies in the South African business environment.

An important aspect of customer service in a B2B context is that sales roles are changing and evolving quickly into relationship management between selling and buying organizations [168]. The quality of the service that suppliers' sales employees provide, therefore, is extremely important, especially when salespeople are fully engaged with their B2B customers. To enjoy continued customer support, salespeople need to formulate and maintain strong relationships with customer procurers, the buyer's management, and support staff [119–121]. The study adds value to social and economic pillars' discourse since it confirmed a positive significant association between the relationship with salespeople and management and preference of suppliers' salespeople. Zolkiewski et al. [169] and Sharma and Dass [170] concur with the findings of this study by stating that relationships formed between the supplier and its customer over the long term are essential, and that building a strategy to increase supplier engagement is the most important job of B2C and B2B marketers as it helps salespeople to connect with customer staff on a personal basis. Ndubisi and Nataraajan [115] assert that creating enduring and sustainable relationships with manufacturers, distributors, and suppliers ensures, for the buying firm, an inflow of quality products and services, and by practicing similar tactics, providers of the service will be able to ensure greater coverage, brand awareness, and profits. Therefore, a sound relationship with salespeople and management has a significant favorable influence on preference of suppliers' salespeople in the new B2B supplier preference model, which partially realizes Mohan's [64] aim to increase B2B buyer behavior research that investigates relationship marketing, organization relationship, and sales and personal selling.

As a result of BEE, it is important for organizations to ensure that their BEE status is high, since the higher the vendor's BEE rating, the more likely it will receive support from large organizations, government departments, and firms that insist that their suppliers are BEE compliant [124,171]. Therefore, it is very important that the salesforce be representative of the "New South Africa," in that the salespeople reflect the requisite diversity. The study makes a unique contribution to social and economic pillars that are inherent to South Africa, since it verified a positive significant relationship between BEE status and preference of suppliers' salespeople that addresses the mandate for B2B procurement and supplier research to executed in divergent contexts [21,22,25,50–52]. EE status has a significant favorable influence on preference of suppliers' salespeople, since if the vendor's salesforce is not representative then the BEE status will not be a genuine endeavor by the vendor to have a salesforce that ignores employing women and other previously disadvantaged people [172]. Finally, organizations' BEE ratings at particular levels improve their prospect of successfully tendering for being a government business, which has the potential to increase their profitability.

Therefore, the study addresses Qazi and Appolloni [27], Salam and Bajaba [61], and Manchanda and Deb's [63] mandate for additional inquiry on SCM, marketing, and B2B procurement operations across multiple industries and/or in developing countries. Furthermore, the newly proposed B2B supplier preference model and the corresponding discussion above should help to lessen the risk in an everchanging and volatile economic climate, especially in developing countries, which also fulfills the request for the development of new B2B procurement models and adds significant value to B2B supplier and procurement discourse [63–67].

## 6. Conclusions

This study concludes that the level of vendor service quality has a positive impact on access to buyer/customer personnel, and by doing so, it can assist the vendor to meet and exceed customer expectations. Hence, vendors obtain customers and also retain them for a long period of time by providing service excellence (i.e., service quality that is reliable and constantly high). The inquiry also concludes that service quality has a significant favorable effect on environmental sustainability in the new B2B supplier preference model. Much worldwide attention focuses on the importance of environmental sustainability and the protection of ecology. The study concludes that B2B procurers that employ EE and AA in

their working environment, and engender a culture of EE, AA, and internal motivation lead to access to vendor personnel. These positive characteristics have a positive effect on the employees of the suppliers, thereby forging a co-dependent relationship between the two entities, which is important for their mutual success. The study concludes that the environmental sustainability of the vendor can have a significant favorable influence on personal relationships if both entities see the value of adopting sustainable environmental practices. Hence, this inquiry provides evidence to prove that the environmental sustainability of the vendor can influence the personal relationship of its B2B customers.

This inquiry concludes that personal relationships and gifts can have a marked positive impact on the relationship between the salesperson, the buyer, and his or her manager if it is given and received in good faith and not at the expense of the buying firm. Personal relationships are naturally important as these relationships can enhance the cooperation between the two entities in the equation. The study concludes that the culture, EE, and AA of the supplier can have a marked impact on both the vendor and the B2B buyer in terms of the adoption and practice of BEE. BEE encourages businesses to integrate people of previously disadvantaged people in the workspace. EE encourages fair representation of members of minority groups, women, or other people, whereas AA involves the inclusion of various groups such as their race, sexuality, faith, gender, and nationality in the workplace. The inquiry concludes that there is a favorable influence of preference of suppliers' salespeople if a sound relationship exists with vendor staff and management. The relationship between the supplier and the buyer's employees will determine the influence that the vendor's salespeople have over the customer's procurement officials and managers. If the relationship is of such a nature that it enhances strong and mutually rewarding bonds between the two companies, then the buyer and seller will benefit from such a union. This inquiry concludes that BEE status has a significant influence on the preference of the suppliers' salespeople. This is because many organizations insist that vendors become BEE compliant, otherwise they will be divorced from bidding for business.

This study addressed a number of research gaps and achieved its main research objective via the development of a new B2B supplier preference model, and by exploring a number of procurement and supplier factors that originated from the 3Ps of the TBL sustainability framework (many of which are unique in South Africa). The inquiry examined new sustainable B2B procurement environmental, social, and economic factors via a large sample that included procurement staff at different levels from an African developing country perspective, which highlighted the importance of procurement and several new 3Ps associations between the above-mentioned factors as an essential sustainable business activity.

### 6.1. Practical Implications

The study provides B2B procurement organizations and supplier companies with many practical implications that emanate from the associations between the new sustainable B2B supplier preference model and supplier constructs. Vendor companies can "develop, implement, and control" a service quality strategy that is focused on forging close, conjointly beneficial, and sustainable relationships with buying organizations (B2B customers) and their employees with the objective of exceeding customer expectations. A service level agreement (SLA) will be beneficial in this regard to spell out the mutual obligations between the B2B procurers and supplier organizations. Furthermore, in an era that encourages 'virtual' contact between the buyers and the suppliers, vendor companies should encourage their employees, and particularly their service staff and salespeople, to obtain as much access to a procurement organization's employees as possible, so that a personalized level of service may be generated that meets their customers' expectations. It is also suggested the vendor companies should be proactive in this regard by procuring eco-friendly offerings and packaging as sustainable inputs for the B2B buying organizations, as well as seeking to instill a culture of sustainability in the procurer organization's workplace by employing salespeople who are mindful of the importance of sustainability. The vendor companies

should also formulate a waste-management strategy, which should be coordinated with the company's procurement and SCM strategies. Lastly, vendor companies should employ lean, green, and agile tactics to service the needs of their B2B buying organizations, thereby inculcating a culture of sustainability.

Effective communication is also a prerequisite for such a coordinated relationship. Therefore, it is proposed that if the B2B buying organization has not adopted EE and AA, then the vendor company should help them to do so. Having access to vendor personnel and allowing them to exceed expectations by means of effective, efficient, and employee-centric service will assist B2B procurement organizations and supplier companies to achieve their goals. Once again there could be an inclusion in the SLA between the B2B buying organizations and vendor companies. To achieve these goals, both supplier companies and B2B procurement organizations require a relationship built upon transparency, mutual trust, and co-dependence as stated above.

The vendor company has a responsibility to ensure that its offerings are green and will not harm the environment or create unnecessary waste. So, by working collectively and synergistically together, the B2B procurement organizations and supplier companies can have a positive impact on the ecology. Hence, it is recommended that the issue of environmental sustainability be addressed as part of the SLA between the B2B buying organization and the vendor company. The SLA should underscore the significance of B2B procurement organization and supplier company contributions and obligations to the practice of environmental sustainability. So, it is further suggested that the procurement officer should be encouraged to form a strong personal and working relationship with vendor company salespeople, and, where appropriate, to give or receive gifts. Although there is controversy about the provision of gifts in a business environment, particularly to procurement professionals, gifting is still seen by many as an expression of gratitude for business received and to forge stronger relationships between the B2B buying organization and the vendor company. The gift, however, must be of such a nature (in quantum or form), that it cannot be perceived to be a bribe. The occasional entertainment of the salesperson will also assist in improving the relationship between the B2B procurement organization and supplier company if it is representative of the level of business or potential customer support.

The study also proposes that culture, EE, and AA should be inculcated in the B2B procurement organization and supplier company so that they will support the adoption and practice of BEE. Therefore, it is recommended that if the vendor company is not BEE compliant, it should endeavor to secure a rating. The vendor company's BEE rating also dictates the attractiveness it offers buyer and seller organizations, so to enhance its chance of bidding for or securing more orders, it should attempt to increase its rating. Those B2B procurers and supplier companies that are fully compliant should flaunt it as a differential advantage over others who are not so inclined. Lastly, it is recommended that the vendor company's salespeople and management team form close and mutually rewarding relationships with its B2B procurement organizations as such relationships will be of benefit to all the stakeholders who are involved with the organizations. Therefore, the study proposes a number of recommendations and suggestions, which originated from the new B2B supplier preference model and corresponding associations to assist buying and supplier organizations in realizing sustainable B2B procurement and vendor objectives in terms of the planet, people, and profit pillars in the African developing economy.

*6.2. Theoretical Implications*

As previously mandated, the study makes an original theoretical contribution in terms of the new sustainable B2B procurement framework, viz., environmental (planet) [21,22,24–26], social (people) [22,25,51,53–59], and economical (profit) factors [23,50] via the development of a new B2B supplier preference model [27,63,65–67] and the assessment of the various procurement/supplier factor associations [21–23,25,51,55,60,61,64] in a multisector and African developing country context [22,25,55,60–63]. The study added to the TBL sustainability

framework and 3P theory by identifying a number of associations between the various planet, people, and profit pillars. The research showed that the association between service quality (profit) and personnel and exceeding expectations (people) was positive, which adds to the TBL framework in terms of these sustainable B2B procurement pillars [83–87,146,147]. Service quality (profit) was also found to positively influence environmental sustainability (planet). Hence, this adds to the body of knowledge regarding the importance of sustainable business strategies and offerings between the B2B and vendors, which should include the 3Ps [148–152]. The research verified positive personnel and exceeding expectations (profit and people) and culture, EE, and AA (people) associations. This makes a theoretical implication to the TBL since there is a dearth of research that considers culture, EE, and AA (uniquely South African supplier factors) in terms of sustainable B2B procurement [153,154]. The study made a further original contribution to theory when it established a positive environmental sustainability (planet) and personal relationships and gifts association (people and profit). Environmental sustainability is of mutual benefit to all stakeholders, which can be facilitated by closer relationships and small gifts. There is a dearth of inquiry that examines parallel sustainability and risk management issues [103], so this result added to the TBL framework and limited sustainable B2B procurement discourse in this regard [155–160].

The research added to the B2B procurement and supplier TBL and 3Ps theory when it established a favorable personal relationships and gifts (people and profit) and salespeople and management (people) association. Personal relationships and gifts can be of significant value to buying and selling organization personnel through effective communication, which ultimately reduces costs and increases profits [161–166]. The positive culture, EE, and AA (people) and BEE (people and profit) association makes a major theoretical contribution in terms of B2B procurement and supplier and the TBL framework since these factors are inherently unique to South Africa [28,32–36]. Another TBL framework theoretical implication was the contribution of the positive relationship with salespeople and management (people) and the preference of suppliers' salespeople (people and profit). Salespeople form strong relationships with the B2B procurers, management, and staff, which motivates continual support and increased profit [119–121,168–170]. One more contribution to knowledge was realized when BEE status (people and profit) was found to positively influence the preference of suppliers' salespeople (people and profit) in the new B2B supplier preference model [31,32,124,171,172]. Organizations receive points, which contribute to their BEE compliance certificate for procurement from Black-owned suppliers. Hence, higher levels of BEE compliance increase the prospect of securing government tenders and other medium to large supplier businesses, which is a unique South African business environment factor [123,124].

Therefore, this inquiry adopted a number of different sustainable procurement and supplier factors that originated from the TBL framework, which included the 3Ps, to develop a new B2B supplier preference model. As discussed above, the associations between these B2B procurement and supplier constructs were not previously considered and/or were unique to the South African business context, so this study further expanded on the TBL and 3Ps theory.

### 6.3. Limitations and Further Inquiry Opportunities

The study is not without limitations. Respondents may have been tempted to provide positive information on many of the environmental and social issues to place their organizations in a positive light. To mitigate this issue, the researchers informed all respondents that the information was completely anonymous and that there would not be mention of the organizations' names in published research outputs. Cross-sectional research was conducted, so further research could monitor the B2B supplier preference factors longitudinally, which would provide more comprehensive results over a longer period. Another limitation is that the research only surveyed the B2B buyers, so future research should also consider the supplier organizations. Additional research could also consider how technology may influence the associations between the various TBL elements that were explored in the

proposed B2B supplier preference model, since this element was not considered in the current study. The study was also limited to a single African developing country, so further inquiry could use the constructs and associations from the B2B supplier preference model that are relevant to other countries across the globe. The research also highlighted that B2B procurement is not a cold affair but is relationship-oriented and more of a personalized endeavor. Hence, research into relationship buying as opposed to relationship selling could also unearth helpful information. The aspect of the marriage between marketing and SCM could also be an interesting topic as can an investigation into the relationship between nepotism, gender, and culture.

**Author Contributions:** Conceptualization, R.D. and M.W.; methodology, R.D. and M.W.; data curation, M.W.; validation, R.D. and M.W.; formal analysis, R.D. and M.W.; investigation, R.D. and M.W.; resources, R.D. and M.W.; supervision, R.D.; writing—original draft preparation, R.D.; writing—review and editing, R.D. and M.W. All authors have read and agreed to the published version of the manuscript.

**Funding:** This research received no external funding.

**Institutional Review Board Statement:** The study was conducted according to the guidelines of the Declaration of Helsinki and approved by the Cape Peninsula University of Technology Research Ethics Committee (FOBREC23082007).

**Informed Consent Statement:** Informed consent was obtained from all subjects involved in the study.

**Data Availability Statement:** The data presented in this study are available upon request from the corresponding author. The data are not publicly available due to restrictions.

**Conflicts of Interest:** The authors declare no conflict of interest.

## Appendix A

**Table A1.** B2B Supplier Preference Codes and Items.

| Construct | Code | Items |
|---|---|---|
| Preference of suppliers' salespeople | PPS1 | I prefer to support a supplier's salesperson that is of the same gender as I. |
| | PPS2 | I prefer to support a vendor's salesperson if he or she is of the same race as I. |
| | PPS3 | I prefer supporting a vendor's salesperson that is roughly my age. |
| | PPS4 | I am strongly influenced to support a vendor's salesperson that speaks to me in my mother tongue (my home language). |
| | PPS5 | I prefer to support a vendor's salesperson that has approximately the same number of years of experience as I. |
| | PPS6 | I prefer the vendor's salesperson to be as educated and as qualified as I. |
| | PPS7 | I prefer to support a vendor's salesperson that has a similar status in his or her organisation as I. |
| | PPS8 | I prefer to support a vendor's salesperson that has the same decision-making latitude as I have. |
| BEE status | BEES1 | A supplier's BEE status is always taken into account before our firm supports such supplier. |
| | BEES2 | The higher the supplier's BEE rating the greater the chance of the supplier getting orders and support from my organisation. |
| | BEES3 | The BEE ratings of my suppliers' suppliers are also taken into account before we as a firm support such an organisation/supplier. |
| Service quality | SQ1 | The quality of the supplier's total service package is also an important consideration when appointing a vendor (e.g., installation, product maintenance, delivery etc.). |
| | SQ2 | A supplier's quick response time to customer queries plays an important role in appointing a supplier. |
| | SQ3 | Supplier flexibility (ability to meet changing customer needs and environments) also plays an important role when selecting vendors. |

**Table A1.** *Cont.*

| Construct | Code | Items |
|---|---|---|
| Relationship with salespeople and management | RSM1 | Having a good relationship with supplier personnel other than its salespeople is a strong motivator when selecting and supporting such a supplier. |
| | RSM2 | Regular communication from a supplier's sales management team and other senior personnel influences whether I support the supplier or not. |
| | RSM3 | Regular constructive visits from sales personnel increase the likelihood of me placing business with the salesperson and his or her employer. |
| Environmental sustainability | ES1 | A supplier with a good track record regarding health, safety and environmental issues is more likely to get our support as opposed to one that does not have such a record. |
| | ES2 | An organisation that markets 'green' environmental-friendly offerings has a better chance of doing business with our firm than one that does not offer such offerings. |
| | ES3 | An organisation that markets products that can be easily disposed of at the end of their lifecycles (bio-degradable products for example) is more likely to be selected as a vendor than one that does not market such products. |
| Culture, EE, and AA | CEEAA1 | The reputation of the vendor regarding the way it treats its employees is an important selection criterion. |
| | CEEAA2 | The culture of the supplier is an important consideration when selecting a vendor. |
| | CEEAA3 | A vending organisation that exercises Employment Equity principles at the workplace is most likely to gain the support of our firm than one that does not. |
| | CEEAA4 | A supplier that exercises Affirmative Action has a better chance of winning the support of my firm that one than does not. |
| Personal relationships and gifts | PRG1 | I try to support salespeople with whom I have a personal relationship with outside of the work environment. |
| | PRG2 | Being regularly entertained by supplier personnel acts as a motivator to support such supplier. |

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
