# Peer review of "Modeling a New Supplier Preference Paradigm: A Business-to-Business and African Developing Economy Context"

_sustainability, doi:10.3390/su15010411_

Round 1
Reviewer 1 Report
I am pleased to have the opportunity to review this research paper. This study attempted to explore a Modelling a New Supplier Preference Paradigm: A Business-to-Business and African Developing Economy Context. Although the topic of this research study is interesting and fits within the journal scope, I think authors should apply the comments indicated below to increase the quality of research justification, contributions and findings. The manuscript know lacks in scientific style and structure.
First of all, paper research gap. Please improve this part in introduction section. Introduction is very general and lacked alignment to the research findings, no discussion was provided to derive the implication from. Theoretical and pragmatics implication are vague and need to be better aligned with this paper theoretical underpinnings and proposed process. Furthermore, there is insufficient support and weak arguments in support of the objective that is proposed as well as the model developed. In the final part of the introduction the objectives proposed, originality and gap that would be better covered. Also how the author will perform the methodology.
the topic of this research study is interesting and fits within the journal scope, I think authors should apply the comments indicated to increase the quality of research justification, contributions and findings
What is the originality of this research? Paper research gap and originality should be better presented at the end of introduction section
Please consider this structure for manuscript final part.
-Discussion
-Conclusion
-Managerial Implication
-Practical/Social Implications
-Discussion needs to be a coherent and cohesive set of arguments that take us beyond this study in particular, and help us see the relevance of what authors have proposed. Authors should create an independent “Discussion” section. Author need to contextualize the findings in the literature, and need to be explicit about the added value of your study towards that literature. Also other studies should be cited to increase the theoretical background of each of the method used. Findings should be contextualized in the literature and should be explicit about the added value of the study towards the literature. Limitations and future research
Questions to be answered:
What practical/professional and academic consequences will this study have for the future of scientific literature (theoretical contributions)?
Why is this study necessary? should make clear arguments to explain what is the originality and value of the proposed model. This should be stated in the final paragraphs of introduction and conclusion sections.
Author Response
Please review to the attached file!

Reviewer 2 Report
Dear authors,
Thank you for the opportunity to review your manuscript.
The purpose of this manuscript is to develop a new supplier preference model by exploring the preference of suppliers’ salespeople „in terms of Black Economic Empowerment (BEE) status; service quality; relationships; environment sustainability; culture, employment equity (EE) and affirmative action (AA); personal relationships and gifts; and access and exceeding expectations”. To achieve this purpose, the authors adopted the quantitative approach to garner the requisite data via a structured questionnaire to survey small, medium, and large-sized organizations operating in South Africa. The data was collected from 445 organizations in a business-to-business (B2B) African context.
In my opinion, the paper addresses a very interesting and topical issue, with a very interesting empirically analysis. The results of this analysis enrich the research on the supplier preference in a developing economy context.
However, from my point of view, some changes are necessary
First, the "Introduction" isn't really an introduction. It should be divided into two separate parts: 1. Introduction and 2. Literature review.
Second, they should mention the limitations of the research.
Overall, I evaluate the study very positively and I recommend its publication.
Author Response
Please review to the attached file!

Reviewer 3 Report
“Modelling a New Supplier Preference Paradigm: A Business-2 to-Business and African Developing Economy Context” article contributes to the field but needs substantial improvement.
1. Abstract must be presented in precise it must include; method result and implications part too.
2. I have not found the literature review in elaborated manner. Authors must include a few more relevant article on environmental sustainability. “Success factors for the adoption of green lean six sigma in healthcare facility: an ISM-MICMAC study” “Green lean six sigma sustainability–oriented project selection and implementation framework for manufacturing industry”, “Green Lean Six Sigma critical barriers: exploration and investigation for improved sustainable performance”, “Grey relational analysis of Green Lean Six Sigma critical success factors for improved organizational performance”. This will enhance the aspects of the sustainability
3. Authors must represent their methodology with clear diagram that will enhance the visibility and understanding of the adopted method in the research. There is no methodology section include the same.
4. Authors must put structure of the article at the end of the introduction section also highlights major research questions.
5. Authors presented results in the nice manner but more elaboration on the discussion part is needed.
6. There is a slew typo and grammatical error in the manuscript. Please fix the same. Improve the referencing styles
7. Compare the results of your study with previous studies of the same nature. Conclusion must represent after effect of the study.
Author Response
Please review to the attached file!

Round 2
Reviewer 1 Report
congratulations, your work is now better, I ask you before the article is published, to better justify the need for the study with literature and better explain the contribution of your study to academia and companies
Author Response
Congratulations, your work is now better, I ask you before the article is published, to better justify the need for the study with literature and better explain the contribution of your study to academia and companies.
Thank you for the positive feedback! We have added literature to justify the need for the study in two places in the introduction section (please refer to the track changes). We have rewritten much of the theoretical and practical (B2B procurement organization and supplier company) implications sections to make the company and academia contributions more explicit/apparent (please refer to the track changes).
Reviewer 3 Report
The authors have made all the required changes in the article. The article is ready for publication.
Author Response
The authors have made all the required changes in the article. The article is ready for publication.
Thank you for the positive feedback!